# Effect of Sperm Selection by Magnetic-Activated Cell Sorting in D-IUI: A Randomized Control Trial

**DOI:** 10.3390/cells11111794

**Published:** 2022-05-30

**Authors:** Cristina González-Ravina, Esther Santamaría-López, Alberto Pacheco, Julia Ramos, Francisco Carranza, Lucía Murria, Ana Ortiz-Vallecillo, Manuel Fernández-Sánchez

**Affiliations:** 1IVI-RMA Seville, Avda. Américo Vespucio 19, 41092 Seville, Spain; cristina.gonzalez@ivirma.com (C.G.-R.); julia.ramos@ivirma.com (J.R.); francisco.carranza@ivirma.com (F.C.); manuel.fernandez@ivirma.com (M.F.-S.); 2IVI Foundation, Instituto de Investigación Sanitaria La Fe (IIS La Fe), 46026 Valencia, Spain; lucia.murria@ivirma.com (L.M.); ana.ortiz@ivirma.com (A.O.-V.); 3Departamento de Biología Molecular e Ingeniería Bioquímica, Universidad Pablo de Olavide, 41013 Seville, Spain; 4IVI-RMA Madrid, Avenida del Talgo 68, 28023 Madrid, Spain; alberto.pacheco@ivirma.com; 5Facultad Ciencias de la Salud, Universidad Alfonso X “El Sabio”, Villanueva de la Cañada, 28016 Madrid, Spain; 6Departamento de Cirugía, Universidad de Sevilla, Avda. Sánchez Pizjuan S/N, 41009 Seville, Spain

**Keywords:** MACS, sperm, sperm selection, IUI, sperm DNA fragmentation

## Abstract

Clinical outcome in assisted reproduction techniques (ARTs) is mainly influenced by the quality of gametes used. It is known that a high percentage of sperm DNA fragmentation (DNAf) decreases the success of ART clinical results. Therefore, techniques such as magnetic-activated cell sorting (MACS) help to improve results in cases of patients with a high percentage of DNAf. Cryopreservation of sperm in donor intrauterine insemination (D-IUI) treatments increases sperm DNAf, so patients using these sperm samples can benefit from using this technique. This prospective randomized national multicenter study analyzed clinical outcomes of 181 D-IUI treatments. MACS was performed after density gradient centrifugation (DGC) in 90 thawed semen donor samples (MACSG), whereas only DGC was performed in 91 thawed semen donor samples (CG). To our knowledge, this is the first study analyzing the effect of MACS on D-IUI cycles. Our results show no significant differences in gestation, live birth, or miscarriage rates between the two groups. We believe that further studies with a larger sample size are needed to evaluate the application of MACS in combination with standard IUI donor sperm preparations in fertility clinics.

## 1. Introduction

Success rates for assisted reproduction (ART) have improved markedly in recent decades; however, despite great advances, it is still far from being 100% effective. From the male point of view, the improvement in the success of ARTs is closely related to the quality of the sperm [1]; sperm abnormality and low sperm count and motility impair the ability of sperm to fertilize the oocyte. Therefore, the design of a new selection tool for functionally normal sperm may be an appropriate strategy to increase the probability of pregnancy during assisted reproduction treatments.

It has been proposed that one of the reasons for this relatively low efficiency is that there is currently no effective methodology to isolate this specific subpopulation of sperm for use in reproductive medicine. This is especially relevant if it is considered that all the techniques encompassed in this context, by definition, elude sperm selection when operating in vivo [2], increasing the risk of fertilizing the oocyte with a defective sperm, which could lead to failures in embryo development. To address this problem, different sperm selection techniques have been developed based on different functional aspects and activation processes that occur during sperm cell capacitation. One of these procedures is magnetic-activated cell sorting (MACS), which is used to positively identify and eliminate apoptotic cells from the ejaculate by identifying externalized phosphatidylserine residues in apoptotic sperm [3]. This technique also reduces the proportion of sperm with fragmented DNA in the ejaculate before using it for ART procedures.

On the other hand, semen cryopreservation has achieved wide recognition as an invaluable medical intervention to preserve male fertility, due to the simplicity and low cost associated with its application. These clinical indications have now expanded to include donor sperm and “backup” storage for infertile men, allowing for a patient-specific approach to the use of frozen sperm [4]. However, despite its clinical advantages and ease of access, human sperm cryopreservation remains an understated technique, with markedly variable suboptimal success rates. These have been largely attributed to the extensive damage induced by cryopreservation, as it is well-known that this process is related to the activation of the caspase pathway in sperm, an increase in sperm DNA fragmentation, and sperm apoptosis [5,6]. In addition, sperm apoptosis has been shown to be associated with poorer sperm motility, morphology, and sperm deformity index scores [6,7,8].

Intrauterine insemination is the oldest method for the treatment of infertility [9], and, even though in vitro fertilization techniques offer higher pregnancy rates, when donor sperm is used in assisted reproduction, intrauterine insemination is still one of the first-choice options in fertility clinics [10,11]. Intrauterine insemination with donor sperm requires the use of frozen semen, so the objective of our work was to analyze whether the use of MACS represents an additional advantage in patients who undergo intrauterine insemination with donor sperm.

## 2. Materials and Methods

### 2.1. Study Design

This prospective randomized national multicenter study was approved by the institutional Ethics Committee Hospital Universitario Virgen Macarena (Seville, Spain), and all patients signed an informed consent form to participate in the study (internal Ethics Committee number 2045). It was registered in Clinical Trials as NCT03659812. A total of 181 intrauterine insemination with sperm donor (D-IUI) treatments were performed at the fertility clinics IVI Sevilla and IVI Madrid. D-IUI was performed after standard sperm capacitation (control group) or after standard sperm capacitation plus sperm selection via MACS (study group).

Inclusion criteria for D-IUI were women with unexplained infertility (couples diagnosed with severe male factor, same-sex couples, and single women with gestational desire), at least one permeable fallopian tube, and normal ovulation. All those patients were included after signing an informed consent form.

### 2.2. Treatment Group Allocation: MACS and Control Group

After semen capacitation and measurement of the sperm concentration and motility, semen samples were randomized into one of the two treatment groups: MACS group (MACSG) or control group (CG). The randomization scheme was previously generated by one of the authors (C.G-R.) using a web tool (http://www.randomization.com; accessed on 25 February 2013), with randomly permuted blocks of four subjects per block. The list was kept in a locked drawer in the administration office, to which the clinical staff who enrolled the participants in the study had no access. Group allocation was requested by telephone. Physicians and patients were blinded to the assigned study intervention for each D-IUI cycle. After D-IUI, they knew the allocation group. CG semen was thawed and capacitated by density gradient centrifugation (DGC), and then D-IUI was performed, whereas in the MACSG, MACS was carried out after DGC and before D-IUI.

### 2.3. Patient Protocol

Patients underwent controlled ovarian stimulation with doses from 50 to 100 IU of recombinant-follicle-stimulating hormone (rFSH) (Puregon^®^. Organon, Madrid, Spain) regulated based on their response from the third day of menstruation and in absence of a dominant follicle (defined as a follicle larger than 10 mm on the second or third day of menstruation) by transvaginal ultrasound scanning (General Electric Voluson Pro-V ultrasound scanner). Once the dominant follicle reached a mean diameter (d1 + d2/2) greater than 18 mm, a dose of 250 mcg of recombinant Human Chorionic Gonadotrophin (hCG) (Ovitrelle^©^ Merck-Serono, Madrid, Spain) was administered subcutaneously to induce ovulation.

Insemination was performed 32–34 h after subcutaneous injection of recombinant HCG, at about 5 mm from the uterine fundus with a Dolphyn^®^ catheter (Lab. Gynetics. Barcelona, Spain), and after it, the patient remained at rest in dorsal decubitus for 10 min.

Sixteen days after the injection of hCG or GnRH analogue, a blood pregnancy test with quantitative determination of the ß fraction of Human Chorionic Gonadotrophin (ß-hCG) was performed. Any value above 5 mIU/mL was considered positive.

All patients with ß-hCG > 80 mIU/mL underwent a transvaginal ultrasound examination (General Electric Voluson Pro-V ultrasound scanner) at around 10 days after blood pregnancy test to assess the size of the gestational sac, yolk vesicle, embryo, and the presence of embryonic cardiac activity.

Patients with ß-hCG greater than 5 mIU/mL but less than 80 mIU/mL underwent a new ß-hCG determination 48 h after the previous one. In case of a decrease in ß-hCG and/or absence of intrauterine gestational sac 10 days after the first blood pregnancy test, gestation was considered nonprogressive.

### 2.4. Study of Semen Quality and DGC

The allocated donor samples were thawed 90 min before the appointment. Semen samples were analyzed after thawing, according to the criteria of the World Health Organization [12]. The variables analyzed in each sample were volume (ml), concentration (mill/mL), motility (%) and sperm morphology. Briefly, spermatozoa concentration and motility were measured using a Makler Chamber (Sefi-Medical Instruments, Haifa, Israel), and morphology was evaluated after Diff-Quik staining by optical microscopy.

Once thawed and analyzed, semen samples were processed following a sperm capacitation protocol by a 3-layer Percoll^®^ density gradient (95–70–45%). First, semen samples were washed 1:2 (Ferticult^TM^ Flushing Medium, FertiPro, Beernem, Belgium) and centrifuged at 300× *g* for 5–10 min. To obtain dilutions of 45%, 70%, and 95%, Sil-Select STOCK^TM^ (Sil-Select STOCK^TM^, FertiPro, Beernem, Belgium) was diluted in Ferticult^TM^ Flushing Medium. Gradient columns were prepared in Falcon tubes by gently layering 1 mL of each solution, starting with the 95% solution at the bottom and followed by the 70% and 45% fractions. Washed samples were layered on top of the columns and processed by centrifuge at 300× *g* for 10–20 min. The recovered 95% pellet was resuspended in 1–2 mL of wash medium and centrifuged at 300× *g* for 5 min to eliminate colloidal particles and finally resuspended in 0.4–0.5 mL of sperm culture medium. This sample was used for an intrauterine insemination (CG) or a MACS selection process (MACSG) before intrauterine insemination.

After capacitation, concentration and motility parameters were re-evaluated, and the sample was randomized to decide whether MACS would be applied, in which case, after performing the magnetic selection technique, the semen parameters were re-analyzed.

### 2.5. MACS Sperm Selection Technique

In the study group (MACSG), MACS protocol was performed following the manufacturer’s instructions (Miltenyi Biotec, Bergisch Gladbach, Germany). According to these instructions, a minimum of 1 × 10^7^ total spermatozoa are needed. This was not a limitation for the study since the semen samples used were donor samples with semen quality (concentration, motility, and morphology) well above the World Health Organization standards of normality.

Briefly, capacitated sperm sample was centrifuged at 300× *g* for 4 min, pellet was resuspended in binding buffer, and cellular suspension was incubated in the dark at room temperature for 15 min with annexin V-conjugated microbeads (100 μL of microbeads for every 10 million spermatozoa). The separation column, which consists of a coated matrix containing iron balls, was placed and fitted in a magnet. The column was rinsed with 1.5 mL of binding buffer. After rinsing, cell suspension was added onto the column, and the eluded fraction (negative fraction) was recovered into a tube. The negative fraction (unlabeled) contained nonapoptotic sperm cells with intact membranes that passed freely through the column, while the positive fraction (labeled) retained in the column contained the apoptotic sperm cells. Lastly, the eluded fraction was centrifuged, and the pellet was resuspended in fresh medium. After performing the MACS protocol, sperm concentration and motility were re-analyzed, and intrauterine insemination was performed. 

### 2.6. Cycle Outcomes

Data were obtained from our clinical database (SIVIS, IVI Digital Information Management Platform). 

Main clinical outcomes for this study included pregnancy rate defined as the number of pregnancies confirmed by ultrasound related to the number of intrauterine insemination cycles; miscarriage rate, which is the number of spontaneous losses of a pregnancy before the 20th week of total intrauterine inseminations, and live-birth rate, defined as the number of newborns related to the total of intrauterine insemination procedures. 

Regarding secondary outcome measures, the following sperm parameters were included: sperm cell concentration, defined as the concentration of spermatozoa in the seminal sample before and after performing the MACS technique; sperm cell motility, such as the proportion of spermatozoa with progressive motility before and after performing the MACS technique; sperm cell morphology, defined as the proportion of spermatozoa with normal morphology before and after performing the MACS technique; sperm cell viability, which is the proportion of live sperm cells before and after performing the MACS technique; and the sperm sample volume, such as the initial volume of the seminal sample obtained after the ejaculation.

### 2.7. Statistical Analysis

Statistical studies were carried out using the Statistical Package for Social Sciences version 22 (SPSS Inc., Chicago, IL, USA) software, where parametric and non-parametric tests were performed to compare the different variables. For descriptive statistics, data are presented as the means and the 95% confidence interval (CI) of their difference. Each comparison generated a different *p* value. In all cases, *p* value ≤ 0.05 was considered significant. Differences between groups were compared using the chi-square test (χ^2^) and U Mann–Whitney test for non-parametric analysis, and the t test for parametric analysis (two-sided). This paper was written according to the updated guidelines for reporting parallel group randomized trials [13].

## 3. Results

Before the start of stimulation, 108 of the 136 potential subjects who met the initial inclusion criteria agreed to participate in the study and signed the study informed consent form. Four of these patients were excluded after signing the study informed consent form because they wished to cease their participation in the study. Thus, 181 treatments were randomly assigned to one of the two groups: 90 treatments to the MACS group and 91 treatments to the CG (Figure 1). Patients’ enrollment took place between November 2017 and April 2018. Regarding patients’ age, mean age in the CG was 36.96 (95% CI: from 36.17 to 37.76) and 35.75 (95% CI: 34.88 to 36.63) (*p* value = 0.055) in the MACSG. 

Characteristics of the seminal sample and sperm functions obtained before and after capacitation are shown in Table 1. We did not find significant differences between the two study groups for any of the variables evaluated under basal conditions. As expected, there were no significant differences found between the study groups after analyzing the semen parameters of the samples after thawing. However, when the pre-insemination sample was analyzed, significant differences were observed in the concentration and percentage of progressive motility. These data are justified by the fact that in the MACSG, the second sperm selection using the MACS technique led to both a decrease in sperm concentration and an increase in the % of progressively motile sperm.

When we analyzed clinical outcomes, we also found no significant differences between the two study groups for the clinical pregnancy rate, miscarriage rate, or live-birth rate (Table 2).

## 4. Discussion

In assisted reproduction cycles, semen quality is vital to obtain optimal clinical results. In addition, in cases where semen is frozen, greater attention should be paid to the selection process, since it has been shown that freezing favors an increase in sperm DNA fragmentation and apoptotic processes [5,6]. Establishing the best preparation method for collection of high-quality spermatozoa is crucial to improve the results of assisted reproduction. Previously, suitable techniques for sperm preparation were density gradient centrifugation and swim up; these methods advantageously select for high-quality spermatozoa, while eliminating seminal plasma, round cells and high-volume defective sperm [14]. In these approaches, there is no guarantee that sperm cells with an early apoptotic state will not enter the sperm collection pellet.

For this reason, it has been proposed in several studies that MACS could contribute to the improvement of ART clinical outcomes [15,16,17,18]. However, over the years, no consensus on whether this technique provides any additional benefit over ARTs has been reached [19]. This could be due to the differences in the study designs and the diversity of the profiles of patients taking part in these research works [15]. Although some advantage in sperm parameters have been reported, it is still inconclusive whether MACS procedure is valuable for clinical practice and in which patients it should be applied. Finally, studies collectively show that the combination of density gradient centrifugation and MACS might provide slight benefits in patients with male factor undergoing ICSI in terms of clinical pregnancy [20,21]. Type of donor or patient and the combination of MACS with other sperm preparation methods could be determinant for improving assisted reproduction outcomes. Regarding IUI treatments, it has been demonstrated that the use of MACS after DGC improves the clinical outcomes for couples with unexplained infertility and repeated assisted reproductive failure [22].

In the present study, density gradient centrifugation was performed prior to MACS in the study group, whereas only conventional sample processing was performed for the control group. Contrary to other research works, the seminal sample was used in an intrauterine insemination protocol and not in an ICSI. Our results show that there were no significant differences in volume, concentration, motility, or morphology between donor sperm samples of the two groups before processing. However, when we analyzed seminal samples after sperm selection techniques, data showed a significant decrease in sperm concentration and a significant increase in sperm progressive motility when MACS was performed after DGC compared to standard processing. The decrease in sperm concentration in MACSG could be explained by the fact that two consecutive selection processes were carried out in this group. These results are in line with the study published by Berteli [21], although in this case a significant decrease in both concentration and motility was observed; at this point, it should be noted that the authors worked with fresh samples as opposed to the frozen samples used in our study, which may explain why the agreement is not complete.

Regarding clinical outcomes, we also did not find significant differences in clinical pregnancy rates when comparing the group of patients who received donor sperm treated with MACS compared to sperm with standard processing for either the miscarriage and live-birth rate, and although the results seem to favor the group who underwent MACS, data follow the general trend, and no significant differences were registered between groups. The absence of statistical variations may be due to the small sample size of the study, so by increasing it, it would be possible to elucidate if, using the MACS protocol, better spermatozoa are obtained as DNA fragmentation is reduced, which may be associated with higher clinical outcomes [17,18,23].

In summary, based on new data and a subsequent deeper understanding of human sperm physiology, innovative advanced sperm selection methods have been developed that go beyond conventional parameters and aim to address more complex and specific cases [24]. Among the limitations of our study, it should be noted that sperm selection through MACS was indicated for a selected population of infertile men with a poor reproductive prognosis and/or high DNA fragmentation index [3,19,25], which differs from our study population that focused on sperm donors characterized by high sperm quality.

## 5. Conclusions

In conclusion, the use of MACS in intrauterine insemination cycles with sperm donors does not seem to provide any additional benefit to the clinical outcomes. Although the use of some of these advanced sperm selection methods has been shown to be able to improve the results of ARTs, most are still in an embryonic state and require further studies to validate their efficacy in clinical practice. These need to include a larger sample size to validate the application of MACS in combination with standard IUI donor sperm preparations in fertility clinics.

## Figures and Tables

**Figure 1 cells-11-01794-f001:**
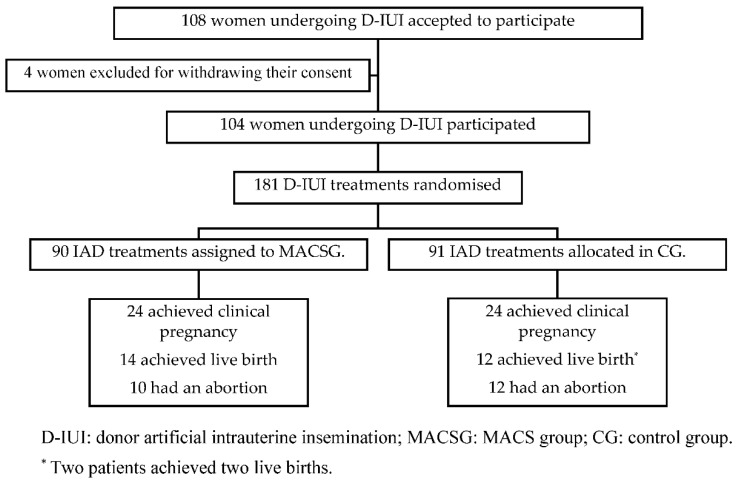
Flowchart of eligible treatments and allocation: MACSG versus CG group.

**Table 1 cells-11-01794-t001:** Seminal parameters. Characteristics of the seminal samples and sperm functions obtained after ejaculation and before insemination in MACSG and CG.

	Basal				Before Insemination			
Parameter (Unit)	MACSG (95% CI)	CG (95% CI)	MD (95% CI)	*p*	MACSG (95% CI)	CG (95% CI)	MD (95% CI)	*p*
Volume (mL)	1.65 (1.55 to 1.75)	1.56 (1.44 to 1.67)	0.10 (−0.06 to 0.25)	0.22	0.44 (0.43 to 0.45)	0.43 (0.42 to 0.44)	0.01 (−0.004 to 0.03)	0.15
Concentration (10^6^ cells/mL)	62.09 (57.96 to 66.23)	64.64 (60.40 to 68.89)	−2.55 (−8.44 to 3.34)	0.40	23.44 (19.84 to 27.04)	36.76 (32.34 to 41.19)	−13.3 (−19.44 to −7.20)	<0.001 *
Progressive motility (%)	39.20 (37.12 to 41.28)	39.30 (37.01 to 41.58)	−0.10 (−3.16 to 2.97)	0.95	86.69 (84.94 to 88.44)	82.51 (80.95 to 84.07)	4.18 (1.82 to 6.55)	<0.001 *
Nonprogressive motility (%)	7.84 (6.61 to 9.08)	8.88 (7.66 to 10.10)	−1.04 (−2.76 to 0.69)	0.24	4.70 (3.74 to 5.67)	5.15 (4.40 to 5.91)	−0.45 (−1.65 to 0.75)	0.46
Morphology (%)	7.08 (6.01 to 8.15)	6.89 (5.84 to 7.94)	0.19 (−1.30 to 1.67)	0.80	N/A	N/A		

MACSG: MACS group; CG: control group; MD: mean difference; CI: confidence interval. * Significant results (<0.05).

**Table 2 cells-11-01794-t002:** Main clinical outcomes. Comparison between clinical outcomes of both semen groups.

	MACSG	CG	RR (95% CI)	*p*
Clinical pregnancy rate (%)	26.7	26.4	1.01 (0.623 to 1.642)	0.96
Live-birth rate (%)	58.3	50	1.17 (0.69 to 1.97)	0.56
Miscarriage rate (%)	41.7	50	0.83 (1.45 to 1.55)	0.56

MACSG: MACS group; CG: control group; MD: mean difference; CI: confidence interval. Significant results (<0.05).

## Data Availability

Not applicable.

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
