# Peer review of "Effect of Sperm Selection by Magnetic-Activated Cell Sorting in D-IUI: A Randomized Control Trial"

_cells, 2022, doi:10.3390/cells11111794_

Round 1

Reviewer 1 Report

In the current study the authors aimed at analyzing the effect of MACS on IUI cycles using donor sperm. The manuscript is clear and well writtem. However some issues shoud be addressed in order to improved the manuscript.

  1. Please change the term "deformity" at line 62
  2. A table with basal characteristics of the enrolled patients should be provided
  3. Minimum semen parameters (volume and/or concentration) necessary for MACS application should be clarified in the methods
  4. The loss of volume after MaCS could represent a limiting factor for its apllication, even in frozen thawed donor sperm cycles. Please further discuss this point
  5. The clinical relevance of sperm DNA fragmentation is still a matter of debate. Please further discuss how MACS would be clinically and routinely apllied in an IVF cycle

Author Response

Comments from Reviewer 1

Comment 1: Please change the term “deformity” at line 62.

Response: We thank Reviewer 1 for this comment. We do not have changed the term “deformity”, but we have changed “rate of deformity” to “sperm deformity index score”. According to Aziz et al. (2007), sperm deformity index (SDI) score is an expression of the quality of sperm morphology and can be calculated by “dividing the total number of deformities observed by the number of sperm that were randomly selected and evaluated irrespective of their morphological normality”.

Comment 2: A table with basal characteristics of the enrolled patients should be provided.

Response: We thank Reviewer 1 for this valuable suggestion. We have included patients’ mean age in each study group in the Results’ section. No other basal characteristics were included since only female patients who met the clinical inclusion criteria for donor IUI were included: women with unexplained infertility (couples diagnosed with severe male factor, same-sex couples and single women with gestational desire), at least one permeable fallopian tube and normal ovulation.

Comment 3: Minimum semen parameters (volume and/or concentration) necessary for MACS application should be clarified in the methods.

Response: We thank Reviewer 1 for this suggestion. We have included in the “2.5 MACS Sperm-Selection Technique” section the total of sperm cells recommended by the manufacturer to start with. Furthermore, we have justified why this minimum was not a limitation in our research project: all the sperm samples used were donor samples with concentration well-above the normal criteria of the World Health Organization.

Comment 4: The loss of volume after MaCS could represent a limiting factor for its application, even in frozen thawed donor sperm cycles. Please further discuss this point.

Response: We thank Reviewer 1 for the comment. The last step of the MACS technique consisted of a centrifugation of the eluded fraction (non-labelled sperm cells = non-apoptotic sperm cells). Then, supernatant was completely aspirated and sperm cells were always resuspended in 0,4-0,5 ml of appropriate medium to perform IUI technique. Therefore, we do not believe that the MACS technique itself is responsible of a loss of volume of the sample. What is more, we detected no significant differences in the volume inseminated between both groups (Table 1).

When assessing sperm concentration, according to the manufacturer of the MACS kit used for this research project (Miltenyi Biotec B.V. & Co. KG, Friedrich-Ebert-Str. 68, 51429 Bergisch Gladbach, Germany), direct magnetic labelling requires a minimal number of washing steps and therefore minimizes cell loss. Nevertheless, we must assume a loss of concentration in these samples when applying density gradient centrifugation followed by MACS since two consecutive selection processes are being carried out. In fact, we observed a significant decrease in sperm concentration. As you suggested in your comment, we have included this in the discussion section.

Comment 5: The clinical relevance of sperm DNA fragmentation is still a matter of debate. Please further discuss how MACS would be clinically and routinely applied in an IVF cycle.

Response: We are very thankful to Reviewer 1 for this comment. We do totally agree with the fact that the clinical relevance of sperm DNA fragmentation is still a matter of debate and that further well design prospective randomized trials are needed. An increase in DNA fragmentation because of freezing and thawing processes has been described in the literature (Paasch et al. (2005), Grunewald et al. (2006)). Based on this fact, we designed this pilot study to detect whether density gradient centrifugation plus MACS performance has some benefits regarding clinical outcomes in those patients undergoing IUI with frozen-thawed donor sperm. To our knowledge, no previous research group have studied this in IUI cycles using donor sperm, although, as we stated in the conclusions section, a larger simple size to validate the application of MACS in combination with standard IUI with donor is needed.

Reviewer 2 Report

In this study, the authors show no significant differences in gestation, live birth, and miscarriage rates after MACS application together with D-IUI. Using of this sperm processing suggests no benefit for semen donor samples characterized by high sperm quality. The limitation of this study is the sample size and no characteristics of the female group. 

I have some comments:

How did you estimate the sample size required for this study?

Do you think that incubation with annexin V or generally sperm processing during MACS can affect the viability and functions of high-quality sperm cells?

Are both groups (MACSG and CG) comparable regarding characteristic as age or health condition of male and female patients?

Table 1: Can you express the confidence interval or standard deviation of the mean for MACSG and CG?

Line 252: I cannot verify reference 22, because this article is not English. You mention studies, but there is only one non-English reference. Can you compare the statement in this paragraph with other articles from international journals?

Author Response

Comments from Reviewer 2

Comment 1: How did you estimate the sample size required for this study?

Response: We thank Reviewer 2 for this comment. This randomized control trial was a pilot study. No other similar articles studying the effect of MACS on clinical outcomes in donor IUI cycles have been published. Therefore, we had no other studies on which we could rely to determine the sample size for this project. Taking into account the number of artificial inseminations with donor sperm performed on average in the clinics where this research project was carried out, a recruitment period of 6 months was established in order to have a sufficient number of cycles available for this pilot study.

Comment 2: Do you think that incubation with annexin V or generally sperm processing during MACS can affect the viability and functions of high-quality sperm cells?

Response: We thank Reviewer 2 for this question. Only phosphatidylserine-exposing sperm cells are magnetically labelled with Annexin V Microbeads and these labelled cells are retained within the MACS column (positive fraction). Intact unlabelled cells (negative fraction), which are the sperm cells that are used for the IUI procedure, flow through the column. According to the manufacturer of the MACS kit used for this research project (Miltenyi Biotec B.V. & Co. KG, Friedrich-Ebert-Str. 68, 51429 Bergisch Gladbach, Germany), MACS® Columns are designed to deliver maximum yield of viable cells. The space between the spheres in the column matrix is about 20 times the size of lymphocytes. This way, columns enable gentle flow of non-labelled cells (no stress, no pressure, sticking or compression).

We do not believe that the magnetic forces and/or the incubation with Annexin V can affect the viability and functions of unlabelled non-apoptotic high-quality sperm cells (negative fraction). Even when studying the positive fraction, according to the manufacturer, thanks to the strong magnetic force used, minimal labelling is required, which has some benefits for cells: no non-specific labelling, no cell activation and no alteration of cell characteristics and functionality.

Source: https://www.miltenyibiotec.com/ES-en/resources/macs-handbook/macs-technologies/cell-separation/magnetic-cell-separation.html#gref

Comment 3: Are both groups (MACSG and CG) comparable regarding characteristic as age or health condition of male and female patients?

Response: We thank Reviewer 2 for this comment. We have included patients’ mean age in each study group in the Results’ section. No other basal characteristics were included since only female patients who met the clinical inclusion criteria for donor IUI were included: women with unexplained infertility (couples diagnosed with severe male factor, same-sex couples and single women with gestational desire), at least one permeable fallopian tube and normal ovulation. Regarding sperm donors, they were under 35 years old and had proven fertility.

Comment 4: Table1: Can you express the confidence interval or standard deviation of the mean for MACSG and CG?

Response: We thank Reviewer 2 for this valuable suggestion and we agree. We have included the confidence interval of the mean for MACSG and CG in Table 1.

Comment 5: Line 252: I cannot verify reference 22, because this article is not English. You mention studies, but there is online one non-English reference. Can you compare the statement in this paragraph with other articles from international journals?

Response: We thank Reviewer 2 for this suggestion. We have tried to find an English reference to compare the statements in this paragraph but we have failed in our search. To our knowledge, this is the only research project that have attempted to demonstrate the benefit of MACS in intrauterine insemination protocols with homologous semen samples. We have decided to remove the paragraph and to include an English reference of an international journal regarding IUI treatments and MACS.

Round 2

Reviewer 1 Report

Although I find it difficult to believe in a clinical application of the current results, the authros addressed the main issues I have raised.

Reviewer 2 Report

The authors have addressed all my comments.